# Environmental variables influence patterns of mammal co-occurrence following introduced predator control

Aurelie M. Kanishka[1]*, Wade Blanchard[1], Tyrone H. Lavery[1,2‡], Natasha M. Robinson[1,3], Nick Dexter[4‡], Chris R. Dickman[5‡], Christopher MacGregor[1], David B. Lindenmayer[1]

1 Fenner School of Environment and Society, The Australian National University, Canberra, ACT, Australia, 2 School of BioSciences, The University of Melbourne, Melbourne, Australia, 3 Conservation and Restoration Science Branch, Science, Economics and Insights Division, NSW Department of Planning and Environment, Australia, 4 Booderee National Park, Jervis Bay Territory, Australia, 5 School of Life and Environmental Sciences, The University of Sydney, Camperdown, NSW, Australia

☯ These authors contributed equally to this work.
‡ THL, ND and CRD also contributed equally to this work.
* aurelie.kanishka@anu.edu.au

**Data Availability Statement:** All csv data files are available from the Dryad digital repository (doi:10.

## Abstract

Co-occurring species often overlap in their use of resources and can interact in complex ways. However, shifts in environmental conditions or resource availability can lead to changes in patterns of species co-occurrence, which may be exacerbated by global escalation of human disturbances to ecosystems, including conservation-directed interventions. We investigated the relative abundance and co-occurrence of two naturally sympatric mammal species following two forms of environmental disturbance: wildfire and introduced predator control. Using 14 years of abundance data from repeat surveys at long-term monitoring sites in south-eastern Australia, we examined the association between a marsupial, the common brushtail possum *Trichosurus vulpecula*, and a co-occurring native rodent, the bush rat *Rattus fuscipes*. We asked: In a fox-controlled environment, are the abundances of common brushtail possums and bush rats affected by environmental disturbance and each other's presence? Using Bayesian regression models, we tested hypotheses that the abundance of each species would vary with changes in environmental and disturbance variables, and that the negative association between bush rats and common brushtail possums was stronger than the association between bush rats and disturbance. Our analyses revealed that bush rat abundance varied greatly in relation to environmental and disturbance variables, whereas common brushtail possums showed relatively limited variation in response to the same variables. There was a negative association between common brushtail possums and bush rats, but this association was weaker than the initial decline and subsequent recovery of bush rats in response to wildfires. Using co-occurrence analysis, we can infer negative relationships in abundance between co-occurring species, but to understand the impacts of such associations, and plan appropriate conservation measures, we require more information on interactions between the species and environmental variables. Co-

5061/dryad.gmsbcc2sh), and have been made public.

**Funding:** The authors received no specific funding for this work.

**Competing interests:** The authors have declared that no competing interests exist.

occurrence can be a powerful and novel method to diagnose threats to communities and understand changes in ecosystem dynamics.

## Introduction

Environmental heterogeneity and inter-species interactions are key factors that influence the composition of ecological communities [1–3]. Communities develop through combinations of species associations and species' niche requirements [3], with niche availability and diversity often influencing the number and identity of species that co-occur [4]. Habitats with high environmental heterogeneity can provide a greater diversity of niche spaces and promote high species richness [3]. Inter-species interactions, such as competition, can similarly influence the assembly of communities and ecological networks, and additionally influence population densities and species dominance [5].

Species co-occurrence analysis is one approach to examine interactions between species in a community. Temporal and spatial co-occurrence patterns are often the product of a mixture of habitat attributes and inter-specific interactions (positive and negative) [6, 7]. Negative interactions between species are likely to reduce their ability to co-occur. However, mechanisms such as environmental and behavioural segregation can mediate negative interactions, enabling co-occurrence, even between highly competitive species [2, 8, 9]. For example, the sympatric seabirds, the common murre *Uria aalge* and the razorbill *Alca torda*, partition their overlapping niches using spatial segregation and differences in foraging behaviour [10]. Similarly, desert rodents in North Africa use separate microhabitats for grazing and protection from predators, which are mediated by the floristic composition of the vegetation [11].

Habitats altered by environmental perturbations can, in turn, affect species' population sizes and ultimately disrupt established patterns of species co-occurrence and community structure. Changes to vegetation structure, for example, have been linked to changes in species diversity [12, 13]. Alterations in fire regimes and fire severity exemplify such impacts [7, 14]. Comparisons of burnt and unburnt sites in central Argentina showed that aggressive ant species became more dominant and competitive in burnt sites compared to unburnt sites [7].

The addition or removal of species from an ecological community can likewise have unexpected impacts on the abundance of other species and trigger trophic or ecological cascades within these communities [15]. The recolonisation of Alaskan kelp forests by sea otters *Enhydra lutris* resulted in heavy depredation of sea urchins *Strogylocentrotus* spp. This caused urchin population declines that, in turn, released kelp from grazing pressure and led to dramatic changes in the composition of the broader kelp forest community [16]. Conversely, the reintroduction of Tasmanian devils *Sarcophilus harrisii* to Maria Island, Australia, resulted in the suppression of feral cats *Felis catus* but also in the unexpected local extinction of the short-tailed shearwater *Ardenna tenuirostris* due to direct predation by Tasmanian devils [17].

The removal of predators (native or exotic) is one particular kind of modification that can lead to profound cascading ecosystem impacts, typically via unmediated increases in the abundance of herbivores [12, 16]. The dramatic population growth of large herbivores, such as deer and macropods, when released from predation, can have damaging effects on animal and plant communities through overgrazing, destruction of habitat, and increased competition [18, 19]. For example, increases in the abundance of Sika deer *Cervus nippon* in Japan altered the composition of dung beetle communities, in turn leading to cascading effects on plant growth, pollination, and ecosystem function [19].

By monitoring patterns of co-occurrence following predator removal, particularly selected pairwise relationships, we can make inferences about the ecological roles of predators, prey

interactions, and cascading impacts on their environments [8]. As many interactions within ecological systems are difficult to observe and extensive datasets are often required to track them, co-occurrence patterns in communities with relatively few species can provide a simpler method of inferring these interactions and allow opportunities to initially observe the effects of environmental change. For example, co-occurrence was higher between two Neotropical marsupials *Didelphis aurita* and *Metachirus nudicaudatus* in Brazilian Atlantic rainforest in the absence of top predators due to apparent competition (where species sharing a common predator show mutual avoidance), which also altered their use of their habitat and resources [20]. Despite the importance of associations between co-occurring species for the maintenance of ecosystem integrity and functioning, how these associations change in relation to environmental change has received relatively limited attention.

In Australia, control of the introduced red fox *Vulpes vulpes*, a common predator of both common brushtail possums and bush rats [21–23], occurs over millions of hectares each year to achieve both conservation and agricultural production objectives [24]. Fox control has been conducted in Booderee National Park, south-eastern Australia, over the past 20 years, and the increase in abundance of common brushtail possums *Trichosurus vulpecula* during the same period is presumed to be a response to this [25]. In contrast, the bush rat *Rattus fuscipes* has declined, but the drivers of this change remain unclear [25]. Booderee National Park also has been subjected to several wildfires in this time, with significant effects on vegetation and temporal activity patterns of small vertebrates [18, 26]. Common brushtail possums and bush rats naturally co-occur, and overlap in their potential diets and patterns of day-to-day activity [27, 28]. A better understanding of associations between bush rats and common brushtail possums might enable us to identify the drivers of potential decline in the former species and the wider implications of invasive predator control (Fig 1).

We asked: 'In a fox-controlled environment, are the abundances of common brushtail possums and bush rats affected by environmental disturbance and each other's presence?' Based on this question, we explored the patterns of occurrence of common brushtail possums and bush rats in response to each other and from the effects of wildfire disturbance using Bayesian regression models by addressing the following hypotheses:

1. Bush rat and common brushtail possum abundances change with variations in environmental conditions and disturbance

2. Bush rat abundance is negatively associated with an increase in common brushtail possum abundance

3. The association with common brushtail possums has a greater effect on bush rat abundance than the association with time since fire disturbance

As we have adopted a correlative method (rather than causative approach) to investigate these patterns, we did not specifically seek to identify the drivers that caused the changes in species abundance that we observed.

## Methods

### Ethics statement

This study was conducted in strict accordance with the recommendations in the *Australian Code for the Care and Use of Animals for Scientific Purposes*. The protocol was approved by the Animal Experimentation Ethics Committee at the Australian National University (Protocol Number: A2021_52).

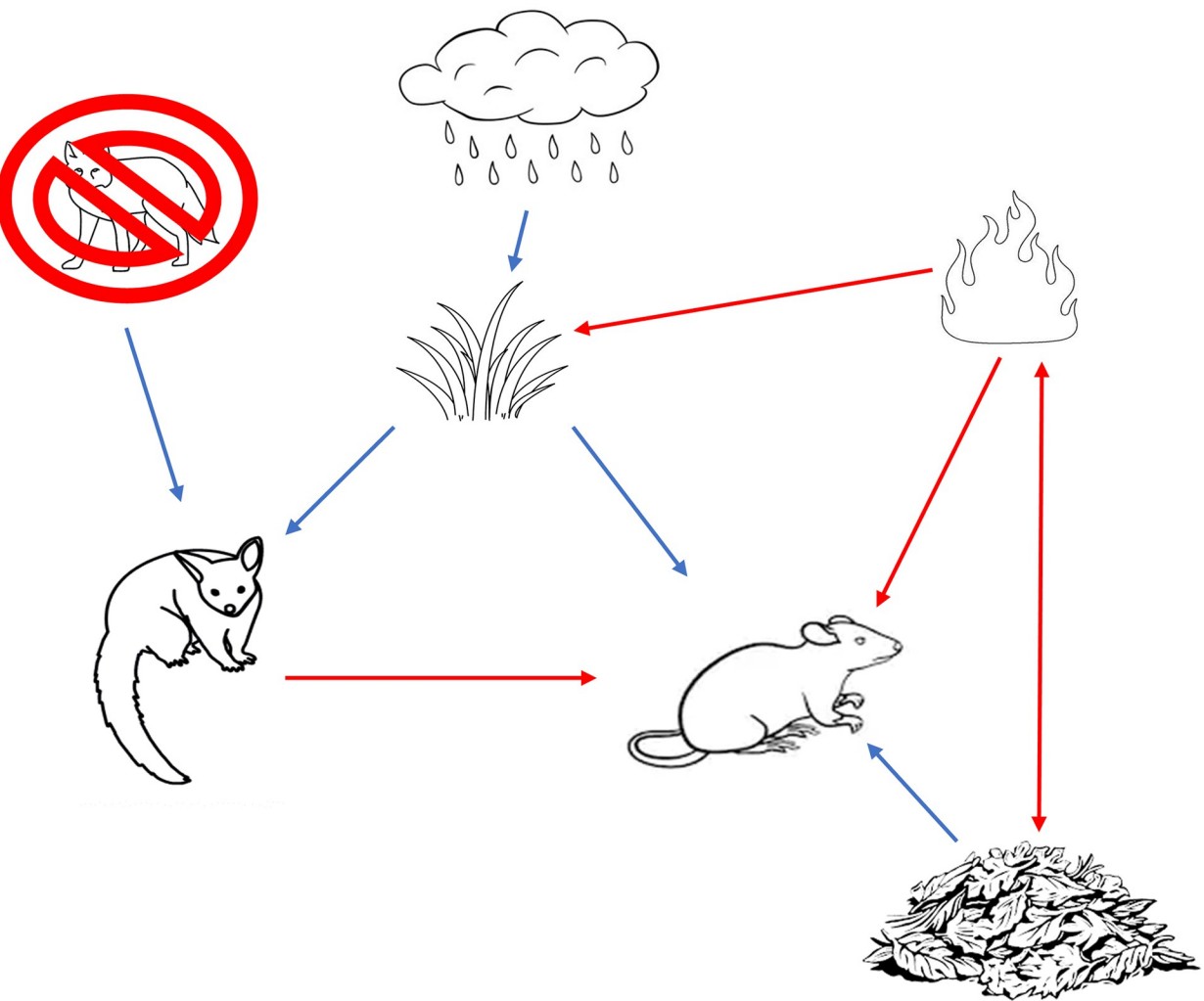

**Fig 1. Conceptual diagram of our predicted ecosystem.** Conceptual diagram of our predictions for how common brushtail possums and bush rats are affected by factors operating within Booderee National Park. Blue lines represent positive relationships, red lines represent negative relationships. Understorey cover is represented with the image of grass, and leaf litter with the image of a pile of leaves.

## Species information

The bush rat is a small, ground-dwelling rodent that ranges from 80 g to 200 g in weight [27]. It prefers habitats with wet, moderately complex vegetation, has a small home range (0.1–0.4 ha), and has limited dispersal ability [27, 29, 30].

The common brushtail possum is a medium-sized partly arboreal marsupial, ranging in size from 1.5 kg to 3.5 kg [31]. It dens in hollows and feeds in the canopy and on the ground [28, 32]. Common brushtail possums show a preference for forest and woodland habitats due to the prevalence of trees with denning hollows in those habitats [28, 32]. It has a relatively large home range, with past research suggesting a variation of 1.7 ha to 12.9 ha dependent on habitat conditions [33].

Both the bush rat and common brushtail possum are nocturnal, generalist omnivores, with a substantial potential overlap in their diets, consuming a large variety of plant matter, fungi, and invertebrates [27, 28].

## Study location

We used data from long-term annual monitoring that commenced in 2003 in Booderee National Park (BNP), Jervis Bay Territory, south-eastern Australia. BNP is on Indigenous land and is jointly managed by the Wreck Bay Aboriginal Community and Parks Australia. The 6600 ha park has a temperate climate, with an average annual rainfall of 1213 mm, spread evenly across the year [34]. Average temperatures range from 18.6–25.1˚C in summer (January) to 9.9–16.1˚C in winter (July) [34]. BNP supports a range of vegetation types such as heathlands, wetlands, forests, and woodlands. Two major fires have occurred in BNP over the past 20 years (in 2003 and 2017), with each burning approximately half of the park. A fox baiting program has been in place in BNP since 1999 and was intensified in 2003 to reduce the deleterious impact of this introduced predator on native prey species [12].

## Data collection

We surveyed small and medium-sized mammals annually each summer for 14 years at 109 permanent sites starting in 2003, with another 20 sites added in 2008. The sites were surveyed along 100 m transects with 2 large (30 x 30 x 60 cm) cage traps at the beginning and end of transects, small (20 x 20 x 50 cm) cage traps every 20 m between the large cage traps, and 10 Elliott traps (10 x 10 x 30 cm; Elliott Scientific Equipment, Upwey, Victoria) every 10 m (Fig 2) [26, 35]. Approximately 50% of the sites were surveyed each year (with the other 50% being surveyed the next year), depending on weather conditions [35]. We recorded the number of individuals of both species caught at each site in a given year.

   We collected data on environmental and disturbance variables at each of our 129 sites. These data included visual estimates of the percentage of understorey and leaf litter cover in four 1 x 1 m subplots within 20 x 20 m survey plots during semi-annual vegetation surveys [26]. We selected understorey and leaf litter as representative variables of the primary bush rat habitat, which are also components of common brushtail possum habitat [27, 28]. We constructed a predictive model to fill the data gaps for those years when sites were not surveyed [37]. We used monthly rainfall averages collected at the nearby Point Perpendicular weather station for the trapping period at each site [34]. We transformed both the vegetation variables and rainfall were transformed into quadratic functions using the poly() function in R [38].

   We used data on fire occurrence recorded on-ground since 2003, and fire history data collected by Booderee National Park over the past 50 years [39], specifically the number of years since the last fire at a site. To minimise possible inaccuracies stemming from incorrect fire dates, or the occurrence of unreported fires, we grouped the number of years since fire into 10-year blocks (i.e., 0–10, 11–20, 21–30, 30+ years).

## Statistical analysis

We used Bayesian regression models with a hurdle step to test the response of species abundances to the selected variables using the brms package ver. 2.16.3 [40–42] implemented in R [38]. These regression models employed Markov Chain Monte Carlo simulations, with four chains and a warm-up of 1000 iterations before sampling another 1000 iterations. We assessed model convergence by ensuring all Rhat values were <1.1 [40, 41]. The hurdle step consisted of two components: the first modelled the presence/absence of the response variable, and the second, conditional on the species being present, modelled the conditional abundance using a zero-truncated Poisson [42]. We combined these two components in an analysis of unconditional abundance [42].

   We created a regression model with bush rats as the response variable, and a regression model with common brushtail possums as the response variable. Both regression models

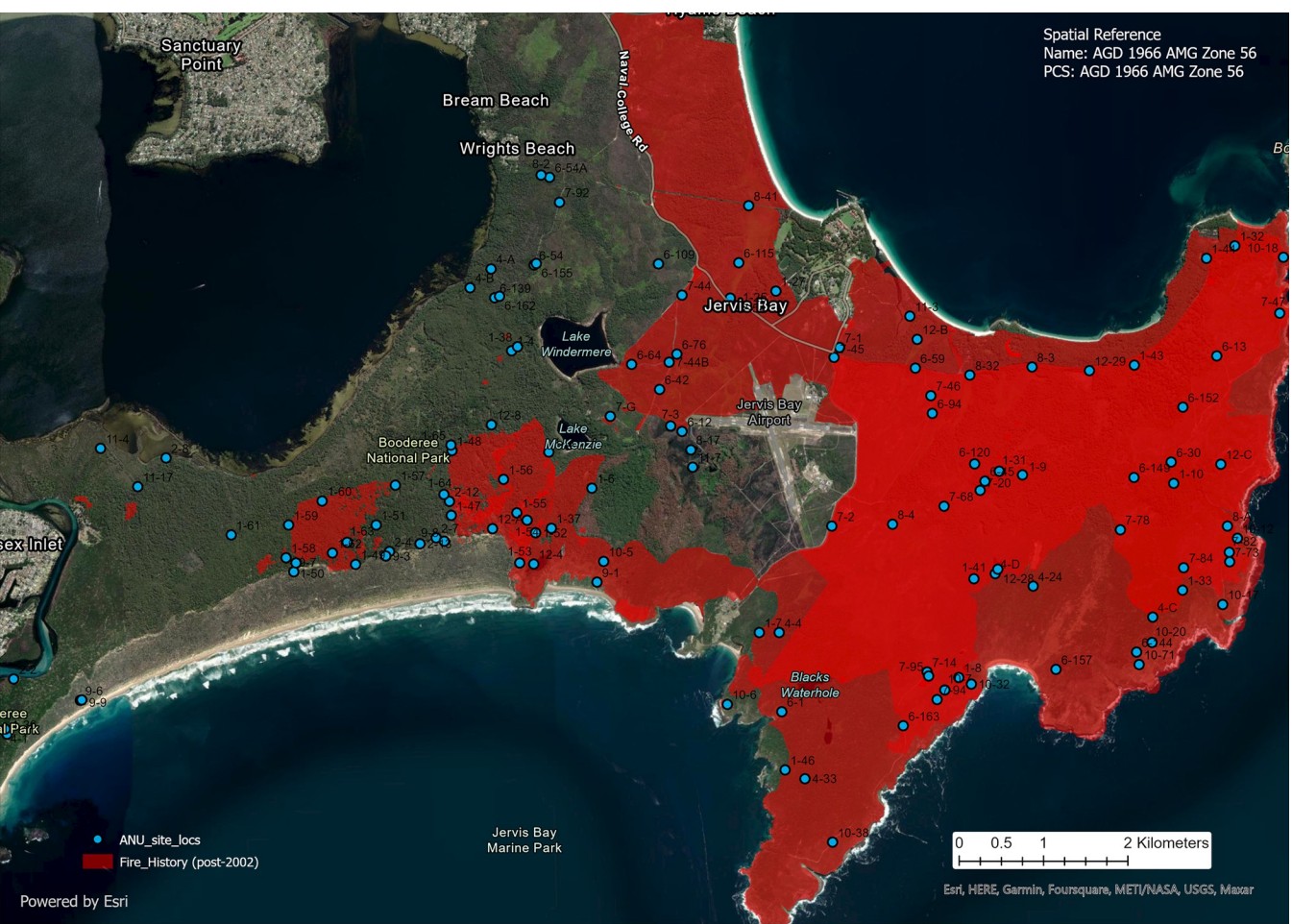

**Fig 2. Locations of trapping sites at Booderee National Park, Jervis Bay Territory, south-eastern Australia.** The red polygon represents areas of the park that were burnt after 2002, with the largest area burnt in December 2003. This map was developed using an ArcGIS Pro basemap, 20 July 2023, Powered by Esri [36] (Sources: Esri, DigitalGlobe, GeoEye, i-cubed, USDA FSA, USGS, AEX, Getmapping, Aerogrid, IGN, IGP, swisstopo, and the GIS User Community) (url: https://cdn.arcgis.com/sharing/rest/content/items/30d6b8271e1849cd9c3042060001f425/resources/styles/root.json).

included time, understorey cover, leaf litter cover and rainfall as covariates within the conditional abundance component. These variables were included to assess the variation in bush rat and common brushtail possum abundances with environmental variables (H1). Years since fire was included in the conditional abundance and hurdle step of both regression models as an explanatory variable, as it is a prominent disturbance within BNP, and past research has indicated that fire has a significant effect on small vertebrate populations [43]. The other species was also input into the conditional abundance and hurdle step of both regression models (i.e., common brushtail possums into the bush rat model, bush rats into the common brushtail possum model) as an explanatory variable to assess the co-occurrence effect between species (H2). We also included site as a random effect, and used the log of the number of Elliott traps as a control for the bush rat models, and the number of open cage traps as a control for the common brushtail possum models. The control variables account for varying trapping effort, and were selected based on the main trap-type that captures the relevant species (i.e., Elliott traps for bush rats, cage traps for common brushtail possums).

We performed a model selection procedure for both of the regression models, based on the selection for explanatory variables only. We chose not to perform model selection on the

covariates (i.e., the environmental variables) as we were testing variation in species abundance in relation to the environment, and not predicting significant changes in abundance that we were with the co-occurrence and disturbance variables. Using model selection, we assessed the relevancy of our exploratory variables to changes in species abundance (H3). The chosen model was the most parsimonious, which was based on the simplest model which was within 2 leave-one-out cross validation (LOOIC) scores of the best fitting model.

We created ten variations of the regression models for each species, and assessed the fit of each variation using LOOIC (Tables 1 and 2) [44]. LOOIC estimates the out-of-sample predictive fit by measuring the predictive accuracy for each data point using a variation of the expected log pointwise predictive density equation [44]. LOOIC was selected as the appropriate method over other model selection methods as it is informative and was created for Bayesian models [44, 45].

## Results

Of the ten bush rat models fitted, the most parsimonious model included an association with years since fire in both the hurdle step and in conditional abundance, while also demonstrating an association with common brushtail possums in conditional abundance (i.e., the change in abundance when species are present; Table 1). Of the ten common brushtail possum model variations fitted, the most parsimonious model included an association with bush rats in both

**Table 1. The bush rat model variations, and associated leave-one-out cross validation (LOOIC) scores, presented in order of best fit.**

| Model | Variables | LOOIC | ΔLOOIC |
|---|---|---|---|
| Model 6 | Year + Rainfall + Understorey + Leaf Litter + Years Since Fire + *T.vulpecula*(A) + Elliotts | 7322.674 | 0 |
| | hu = Year + Years Since Fire + *T.vulpecula*(PA) + Elliotts | | |
| Model 2 | Year + Rainfall + Understorey + Leaf Litter + Years Since Fire + *T.vulpecula*(A) + Elliotts | 7323.115 | 0.441 |
| | hu = Year + Years Since Fire + *T.vulpecula*(A) + Elliotts | | |
| Model 4 | Year + Rainfall + Understorey + Leaf Litter + Years Since Fire + *T.vulpecula*(A) + Elliotts | 7323.418 | 0.744 |
| | hu = Year + Years Since Fire + Elliotts | | |
| Model 9 | Year + Rainfall + Understorey + Leaf Litter + Years Since Fire + *T.vulpecula*(A) + Elliotts | 7332.744 | 10.07 |
| | hu = Year + *T.vulpecula*(A) + Elliotts | | |
| Model 8 | Year + Rainfall + Understorey + Leaf Litter + *T.vulpecula*(A) + Elliotts | 7341.191 | 18.517 |
| | hu = Year + Years Since Fire + *T.vulpecula*(A) + Elliotts | | |
| Model 3 | Year + Rainfall + Understorey + Leaf Litter + Years Since Fire + Elliotts | 7344.454 | 21.78 |
| | hu = Year + Years Since Fire + *T.vulpecula*(A) + Elliotts | | |
| Model 5 | Year + Rainfall + Understorey + Leaf Litter + Years Since Fire + Elliotts | 7346.091 | 23.417 |
| | hu = Year + Years Since Fire + Elliotts | | |
| Model 10 | Year + Rainfall + Understorey + Leaf Litter + *T.vulpecula*(A) + Elliotts | 7349.877 | 27.203 |
| | hu = Year + *T.vulpecula*(A) + Elliotts | | |
| Model 7 | Year + Rainfall + Understorey + Leaf Litter + Years Since Fire + Elliotts | 7350.136 | 27.462 |
| | hu = Year + Years Since Fire + *T.vulpecula*(PA) + Elliotts | | |
| Model 1 | Year + Rainfall + Understorey + Leaf Litter + Elliotts | 7371.065 | 48.391 |
| | hu = Year + Elliotts | | |
| Null Model | 1 | 8826.685 | 1504.01 |
| | hu = 1 | | |

The delta score (ΔLOOIC) in the final column represents the overall change in LOOIC score from the best fitting model. The model variations are presented with the first line variables inputted to model conditional abundance, and the second line (starting with hu =) inputted into the hurdle step. Common brushtail possums (written as *T. vulpecula*) were inputted in the model using either abundance (represented with (A)) or as the presence/absence (represented with (PA)). Model 4 (where common brushtail possums were only inputted into the conditional abundance step) was the most parsimonious model, and therefore used in this study.

**Table 2. The common brushtail possum model variations, and the associated leave-one-out cross validation (LOOIC) scores, presented in order of best fit.**

| Model | Variables | LOOIC | ΔLOOIC |
|---|---|---|---|
| Model 8 | Year + Rainfall + Understorey + Leaf Litter + *R.fuscipes*(A) + Elliotts | 1695.8 | 0 |
| | hu = Year + Years Since Fire + *R.fuscipes*(A) + Elliotts | | |
| Model 4 | Year + Rainfall + Understorey + Leaf Litter + Years Since Fire + *R.fuscipes*(A) + Elliotts | 1697.407 | 1.607 |
| | hu = Year + Years Since Fire + Elliotts | | |
| Model 10 | Year + Rainfall + Understorey + Leaf Litter + *R.fuscipes*(A) + Elliotts | 1697.537 | 1.737 |
| | hu = Year + *R.fuscipes*(A) + Elliotts | | |
| Model 6 | Year + Rainfall + Understorey + Leaf Litter + Years Since Fire + *R.fuscipes*(A) + Elliotts | 1699.235 | 3.435 |
| | hu = Year + Years Since Fire + *R.fuscipes*(PA) + Elliotts | | |
| Model 9 | Year + Rainfall + Understorey + Leaf Litter + Years Since Fire + *R.fuscipes*(A) + Elliotts | 1699.804 | 4.004 |
| | hu = Year + *R.fuscipes*(A) + Elliotts | | |
| Model 2 | Year + Rainfall + Understorey + Leaf Litter + Years Since Fire + *R.fuscipes*(A) + Elliotts | 1700.491 | 4.691 |
| | hu = Year + Years Since Fire + *R.fuscipes*(A) + Elliotts | | |
| Model 5 | Year + Rainfall + Understorey + Leaf Litter + Years Since Fire + Elliotts | 1704.973 | 9.173 |
| | hu = Year + Years Since Fire + Elliotts | | |
| Model 3 | Year + Rainfall + Understorey + Leaf Litter + Years Since Fire + Elliotts | 1706.319 | 10.519 |
| | hu = Year + Years Since Fire + *R.fuscipes*(A) + Elliotts | | |
| Model 7 | Year + Rainfall + Understorey + Leaf Litter + Years Since Fire + Elliotts | 1707.678 | 11.878 |
| | hu = Year + Years Since Fire + *R.fuscipes*(PA) + Elliotts | | |
| Model 1 | Year + Rainfall + Understorey + Leaf Litter + Elliotts | 1709.066 | 13.266 |
| | hu = Year + Elliotts | | |
| Null Model | 1 | 1765.325 | 69.525 |
| | hu = 1 | | |

The delta score (ΔLOOIC) in the final column represents the overall change in LOOIC score from the best fitting model. The model variations are presented with the first line variables inputted to model conditional abundance, and the second line (starting with hu =) inputted into the hurdle step. Bush rats (written as *R. fuscipes*) was inputted into the model using either abundance (represented as (A)) or as presence/absence (represented as (PA)). Model 10 (which modelled bush rats and common brushtail possums in both the conditional abundance and hurdle step, but did not model years since fire) was selected the most parsimonious model, and therefore used in this study.

the hurdle step and in conditional abundance, but did not contain an association with years since fire (Table 2).

Our first hypothesis (H1) was that bush rat and common brushtail possum abundances would vary with changes in environmental and disturbance variables. We found that bush rat presence and abundance decreased over time (Table 3), whereas common brushtail possum presence increased over time, but abundance did not (Table 4). The lowest abundance of bush rats was in sites characterised by high and low percentages of leaf litter cover, approximately 50% understorey cover, in periods with high rainfall (Table 3). In contrast, the abundance of common brushtail possums did not vary significantly in response to the selected environmental variables, and the results also demonstrated large variability (Table 4). Bush rat presence and abundance were lowest in the first ten years following a fire with subsequent increases in the following year blocks (Fig 3, Table 3). Conversely, years since fire was found to not be a relevant variable for common brushtail possums, based on the LOOIC scores.

Our second hypothesis (H2) was that bush rat abundance would be negatively associated with an increase in common brushtail possum abundance. Our model demonstrated that, in response to the increasing abundance of common brushtail possums, bush rat abundance declined (Fig 4, Table 3). In the common brushtail possum model, our results demonstrated a

**Table 3. The results of the Bayesian regression for bush rat model 4 (which was selected as the most parsimonious model using the LOOIC score).**

| Variable | m | SE | Lower 95% CI | Upper 95% CI |
|---|---|---|---|---|
| Conditional Abundance | | | | |
| Year | -0.09 | 0.01 | -0.10 | -0.08 |
| Rainfall$_{lower}$ | -2.58 | 0.48 | -3.53 | -1.65 |
| Rainfall$_{upper}$ | 0.33 | 0.48 | -0.61 | 1.25 |
| Understorey$_{lower}$ | -3.91 | 1.30 | -6.53 | -1.40 |
| Understorey$_{upper}$ | 4.68 | 1.05 | 2.57 | 6.73 |
| Leaf Litter$_{lower}$ | 0.66 | 1.82 | -2.85 | 4.28 |
| Leaf Litter$_{upper}$ | -10.33 | 1.22 | -12.75 | -7.89 |
| Years since fire$_{0-10}$ | -0.41 | 0.12 | -0.65 | -0.17 |
| Years since fire$_{11-20}$ | -0.15 | 0.11 | -0.37 | 0.07 |
| Years since fire$_{21-30}$ | -0.13 | 0.09 | -0.31 | 0.04 |
| *T. vulpecula* | -0.08 | 0.02 | -0.11 | -0.05 |
| Elliott | 0.66 | 0.04 | 0.59 | 0.74 |
| Hurdle effects | | | | |
| Year | 0.21 | 0.02 | 0.16 | 0.26 |
| Years since fire$_{0-10}$ | 1.29 | 0.48 | 0.36 | 2.26 |
| Years since fire$_{11-20}$ | 0.44 | 0.49 | -0.50 | 1.45 |
| Years since fire$_{21-30}$ | -0.73 | 0.81 | -2.41 | 0.78 |
| Elliott | -1.07 | 0.24 | -1.56 | -0.60 |

m represents the posterior mean, SE represents the posterior standard error, and the lower and upper 95% CIs represent the lower and upper ranges of the 95% credible interval. 30+ years since fire is not presented in the table because it is assumed not to have a significant effect compared to the other years-since-fire categories [40]. The results of the environmental variables (rainfall, understorey, and leaf litter cover) are quadratic effects. The results of the hurdle effects represent the probability that there are zero bush rats.

**Table 4. The results of the Bayesian regression for common brushtail possum model 10 (which was selected as the most parsimonious model using the LOOIC scores).**

| Variable | m | SE | Lower 95% CI | Upper 95% CI |
|---|---|---|---|---|
| Conditional abundance | | | | |
| Year | 0.01 | 0.02 | -0.04 | 0.05 |
| Rainfall$_{lower}$ | 0.97 | 2.44 | -3.85 | 5.73 |
| Rainfall$_{upper}$ | -1.13 | 2.90 | -6.90 | 4.48 |
| Understorey$_{lower}$ | -12.03 | 4.69 | -21.15 | -2.89 |
| Understorey$_{upper}$ | -3.55 | 4.02 | -11.50 | 4.35 |
| Leaf Litter$_{lower}$ | -5.10 | 5.64 | -15.69 | 6.02 |
| Leaf Litter$_{upper}$ | 4.56 | 3.44 | -2.31 | 11.38 |
| *R. fuscipes* | -0.06 | 0.02 | -0.10 | -0.02 |
| Cages | 0.03 | 0.01 | 0.01 | 0.04 |
| Hurdle | | | | |
| Year | -0.17 | 0.03 | -0.23 | -0.12 |
| *R. fuscipes* | 0.01 | 0.02 | -0.03 | 0.06 |
| Cages | -0.04 | 0.01 | -0.06 | -0.01 |

m represents the posterior mean, SE represents the posterior standard error, and the lower and upper 95% CIs represent the lower and upper ranges of the 95% credible interval. The results of the environmental variables (rainfall, understorey, and leaf litter cover) are quadratic effects. The results of the hurdle effects represent the probability that there are zero common brushtail possums.

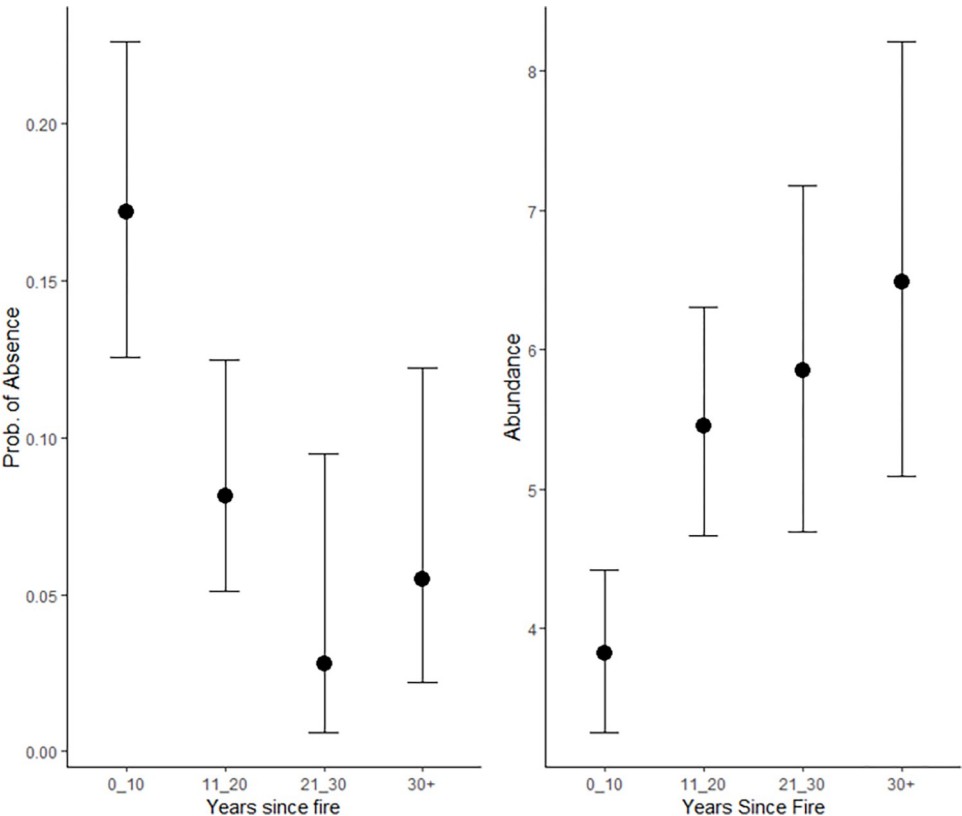

**Fig 3. The associated change in the absence and abundance of bush rats in response to years since fire.** The figure on the left shows the change in probability of bush rats being absent, with data taken from the hurdle step of the model. The figure on the right shows the change in the unconditional abundance. Error bars represent the 95% credible intervals.

similar negative association between common brushtail possums and bush rats (Table 4, S3 Fig).

Our third hypothesis (H3) was that the association between bush rats and common brushtail possums would be more relevant than the association between bush rats and years since fire. Using the LOOIC scores, the model containing years since fire only was a better fit to our data than the model containing common brushtail possums only (Table 2).

## Discussion

Species interactions and associations play important roles in influencing temporal and spatial co-occurrence and can ultimately shape the assembly of biotic communities [1, 6]. However, alterations to the environment can cause fundamental shifts in these associations and interactions with flow-on effects for the presence and abundance of individual species [14, 16]. In this study, we predicted that increased abundance of common brushtail possums would be associated with the decline of bush rats in Booderee National Park. To investigate this, we constructed three hypotheses about links between changes in environmental conditions and species abundances. We firstly investigated the role that changes in environmental conditions have on the abundance of bush rats and common brushtail possums (H1), with our results demonstrating that environmental conditions and the time since fire have large effects on the abundance of bush rats but not that of common brushtail possums.

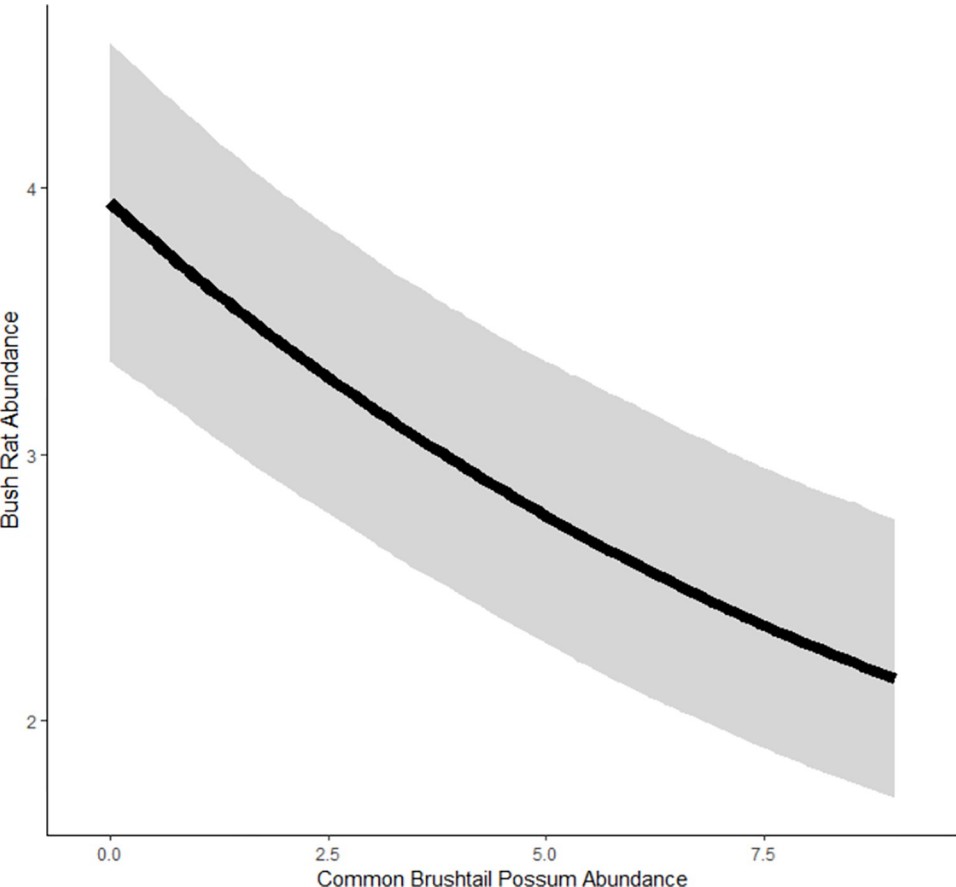

**Fig 4. The unconditional change in the abundance of bush rats in response to the increasing abundance of common brushtail possums.** The shaded regions represent the 95% credible intervals.

We then investigated the strength of a negative association between common brushtail possums and bush rats (H2) and compared this association with that of the influence of time since fire (H3). Our results demonstrate that an increase in the abundance of common brushtail possums was associated with a reduction in bush rat abundance, with this relationship being observed in both bush rat and common brushtail possum models. However, our results did not support our third hypothesis. The decline of bush rats was better explained by our model containing years since fire only compared to our model containing common brushtail possums only. Collectively, our results suggest that while a negative association between common brushtail possums and bush rats exists, the decline of bush rats might be better explained by the influences of environmental and disturbance variables than by any interactions between the two species.

The negative association between common brushtail possums and bush rats could stem from two possible scenarios. First, the abundances of bush rats and common brushtail possums may be unrelated, with each shaped by individual species' habitat and/or condition preferences [46, 47]. A common misstep with co-occurrence models is to assume that the associations between species are indicative of a species interaction, when the models could be producing signals from how species are interacting differently or similarly to environmental conditions [48]. Our results do show that understorey cover and time since fire are both strong indicators of variations in bush rat abundance, while only understorey cover was a weak

indicator for common brushtail possums (Tables 3 and 4). Furthermore, research has demonstrated that disturbances like fire affect the two species to different extents. Bush rats have been recorded to be severely restricted by fire events and the associated habitat changes [26, 43], whereas the effect on common brushtail possums varies between minor negative influence to positive recoveries post-fire [26, 49]. Therefore, despite their extensive co-occurrence and overlap in some food and habitat resources, bush rats and common brushtail possums each most likely exploit different parts of these resource axes, and also exhibit different home range sizes and dispersal abilities (Tables 3 and 4) [27, 32].

Alternatively, a negative correlation between the abundances of common brushtail possum and bush rats could be a result of direct, negative interactions between the two species. While recent research has documented some predatory behaviours in common brushtail possums [17], we predict this negative relationship to be competitive due to the potential overlaps in habitat use, and the greater body of research documenting common brushtail possums' competitive behaviours [27, 28, 50]. Ruscoe and associates [50] demonstrated that common brushtail possums in New Zealand are competitively dominant to smaller mammals by documenting the competitive release of black rats *Rattus rattus* (a species of the same genus and with a similar trophic role as the bush rat) after the removal of possums. However, in contrast to our study, both these species are exotic to New Zealand. We further argue that the level of co-occurrence and potential competition between bush rats and common brushtail possums is likely mediated by habitat conditions as well as environmental conditions (e.g., wildfire events, amount of rainfall).

Changes in species composition or relative abundance can have both direct and cascading influences on community structure [5, 12, 15]. Evidence for this has been derived primarily from complete species removals or additions to communities but has also been deduced from studies of overabundance of herbivores [15, 19]. Introduced species can often increase levels of competition for resources, such as limited nesting hollows by birds, resulting in lower breeding success for native species [51]. Many Australian mammals have been negatively impacted by introduced competitors (e.g., rabbits, deer) and predators (e.g., feral cats, red foxes), and these impacts have been documented to be among the leading causes of declines [12, 52]. Range shifts can likewise increase competitive encounters. For example, the range expansion of barred owls *Strix varia* has increased interference competition with northern spotted owls *Strix occidentalis caurina*, to the latter's disadvantage [53].

Our study highlights potential cascading effects from the loss of an introduced predator [12, 16]. The increased abundance of common brushtail possums is presumed to be a response to the deliberate population reduction of the red fox [12]. The negative correlation in the abundance of bush rats with common brushtail possums may be the product of resulting increased competition, especially given that common brushtail possums are possibly one of the main mammalian competitors [28, 50]. Comparable results have been seen in manipulative field experiments whereby excluding a predator like the shorebird *Calidris pusilla* can have effects that cascade down through different trophic levels with implications for community structure [54].

The fox baiting program in BNP had effects that have cascaded through the park, many of which are not accounted for [25]. There is strong evidence that the removal of foxes has allowed common brushtail possum and macropod populations to increase [12, 25]. There is subsequent evidence that the increased macropod populations have altered vegetation structure, and potentially altered the fire regime (i.e., frequency/intensity), which consequently affects the animal community, as supported by our model [12, 18]. Our model additionally supports the prediction that the declines of small mammals are related to the increases of common brushtail possum, and evidence suggests a similar response would result from increased

macropod abundances [12]. These results demonstrate that an understanding of interspecific interactions is important for successful conservation. As in our study, interspecific interactions also had an influence on the semi-successful reintroduction of sea otters *Enhydra lutris* to the Canadian Pacific coastline [55]. The unsuccessful reintroductions arose from the unforeseen apparent competition with pinnipeds, which occupied areas of good quality foraging habitat to the detriment of otters [55]. Conversely, using the information on shared historical occupancy with the pine marten *Martes martes*, the restoration of this predator is being used to recover red squirrel *Sciurus vulgaris* populations through its role in controlling the invasive grey squirrel *Sciurus carolinensis* in the UK [56].

## Conclusion

An increased abundance of common brushtail possums was associated with a reduced abundance of bush rats. Our results indicate that species presence and interactions can have an important influence on species persistence. However, the strength of interactions may either be moderated by environmental effects, or be more indicative of the different effects of environmental conditions on the two species and therefore managers must take both into account if the management of populations and ecological communities is to be effective. Furthermore, co-occurrence analysis can highlight unknown and potentially detrimental associations, which can be used as a starting point when diagnosing threats to community assemblages. Using co-occurrence models is one way to identify potential negative relationships between co-occurring species after an environmental change and/or following the population growth of one species. This information should be used to direct investigations into the impacts of such associations, by investigating the direct and indirect interactions between co-occurring species.

## Supporting information

**S1 Fig. The forest plot of bush rat model 4.** The blue vertical line represents the zero-effect line. Red horizontal lines represent negative effects, blue horizontal lines represent positive effects. The black dot represents the posterior mean. Lines that cross the zero-effect line represent non-significant results.
(TIF)

**S2 Fig. The forest plot of common brushtail possum model 10.** The blue vertical line represents the zero-effect line. Red horizontal lines represent negative effects, blue horizontal lines represent positive effects. The black dot represents the posterior mean. Lines that cross the zero-effect line represent non-significant results.
(TIF)

**S3 Fig. The unconditional change in the abundance of common brushtail possums in response to the increasing abundance of bush rats.** The shaded regions represent the 95% credible intervals.
(TIF)

## Acknowledgments

We acknowledge and thank the Wreck Bay Aboriginal Community, owners of Booderee National Park for providing access to and support for our work in the park, and to the staff at Booderee National Park and all volunteers for assisting in data collection. We also

acknowledge Dr. Maldwyn John Evans of the Fenner School of Environment and Society, Australian National University, for his support in revising statistical analysis methodology.

## Author Contributions

**Conceptualization:** Aurelie M. Kanishka, Wade Blanchard, Tyrone H. Lavery, Natasha M. Robinson, Nick Dexter, Chris R. Dickman, Christopher MacGregor, David B. Lindenmayer.

**Data curation:** Natasha M. Robinson, Christopher MacGregor, David B. Lindenmayer.

**Formal analysis:** Aurelie M. Kanishka, Wade Blanchard.

**Investigation:** Aurelie M. Kanishka.

**Methodology:** Aurelie M. Kanishka, Wade Blanchard, Christopher MacGregor.

**Supervision:** Tyrone H. Lavery, Natasha M. Robinson, Nick Dexter, Chris R. Dickman, David B. Lindenmayer.

**Writing – original draft:** Aurelie M. Kanishka.

**Writing – review & editing:** Aurelie M. Kanishka, Wade Blanchard, Tyrone H. Lavery, Natasha M. Robinson, Nick Dexter, Chris R. Dickman, Christopher MacGregor, David B. Lindenmayer.

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
