## [Decision Letter · Decision Letter 0]

27 Jun 2023

PONE-D-23-02032Environmental variables influence patterns of mammal co-occurrence following introduced predator controlPLOS ONE

Dear Dr. Kanishka,

Thank you for submitting your manuscript to PLOS ONE. After careful consideration, we feel that it has merit but does not fully meet PLOS ONE’s publication criteria as it currently stands. Therefore, we invite you to submit a revised version of the manuscript that addresses the points raised during the review process.

We look forward to receiving your revised manuscript.

Kind regards,

Daniel de Paiva Silva, Ph.D.

Academic Editor

PLOS ONE

Journal Requirements:

3. We note that Figure 1 in your submission contain map images which may be copyrighted. All PLOS content is published under the Creative Commons Attribution License (CC BY 4.0), which means that the manuscript, images, and Supporting Information files will be freely available online, and any third party is permitted to access, download, copy, distribute, and use these materials in any way, even commercially, with proper attribution. For these reasons, we cannot publish previously copyrighted maps or satellite images created using proprietary data, such as Google software (Google Maps, Street View, and Earth). For more information, see our copyright guidelines: http://journals.plos.org/plosone/s/licenses-and-copyright.

Additional Editor Comments:

Dear Dr. Kanishka,

After this first review round, your manuscript has received three decisions: one minor review and two major reviews. I tend to agree with the reviewers who indicated that more in-deep improvements are required to make your manuscript worthy of publication. Therefore, I invite you to resubmit it with the improvements added to your study. In case you do not agree with any of the issues raised, please do not forget to justify this in the rebuttal letter you will resubmit along with your study. I will grant you a three-month period to deliver the revised version of your manuscript. Please do not hesitate to write me if you need.

Best regards,

Daniel Silva, PhD.

Reviewers' comments:

Reviewer's Responses to Questions

**Comments to the Author**

1. Is the manuscript technically sound, and do the data support the conclusions?

Reviewer #1: Yes

Reviewer #2: Yes

Reviewer #3: Partly

2. Has the statistical analysis been performed appropriately and rigorously? 

Reviewer #1: I Don't Know

Reviewer #2: Yes

Reviewer #3: I Don't Know

3. Have the authors made all data underlying the findings in their manuscript fully available?

Reviewer #1: Yes

Reviewer #2: Yes

Reviewer #3: Yes

4. Is the manuscript presented in an intelligible fashion and written in standard English?

Reviewer #1: Yes

Reviewer #2: Yes

Reviewer #3: Yes

5. Review Comments to the Author

Reviewer #1: Thank you for the opportunity of reviewing this manuscript. An interest study on the variations in the environment and the co-occurrence of small mammal species in Australia. In the comments below, I ask some general and specific questions about the manuscript that authors may consider before publishing. I hope that the comments can be useful and I apologize if some of them were made due to a lack of correct interpretation on my part. I will be happy to review an improved version of the manuscript if requested by the Editors.

Major comments:

Considering the manuscript, I believe the main potential would be how environmental and perturbation variables modulate the co-occurrence and abundance of opossum and rats species. I therefore ask reviewers to first review a new strategy in using the “introduced predator” question, what is discussed in the article are environmental factors influencing species co-occurrence patterns, but I do not see clarity on how the “introduced predator” factor is affecting or influencing the results. I believe that this is a determining factor within the context of the research, but I do not see it being tested or considered in the analyses, I only see it being mentioned.

I note that there was a very intense sample collection effort and I agree with the authors that more information is needed on the interactions between species and environmental variables. Therefore I suggest a new analysis taking into account the landscape of the study area. See the possibility of using images or a landscape database to use a delimitation in your sampling sites (buffer? investigating a minimum area that can be related to the home range of the species in focus), and extract information about this delimitation that may be fundamental in the occurrence of species (percentage of forested area, other natural conformations, distance from water bodies, other classifications). If the delimited areas overlap, randomize the points. Perform correlation analysis between variables and build models to explain how environmental variables and disturbance affect species abundance and co-occurrence. After the modifications, I suggest reassessing the data analysis. I suggest that a table be made in supplementary material with all collection points including values for each tested variable.

Minor issues:

Line 49 - Introduction: View a concept map strategy for getting started. 1. Environmental heterogeneity and factors that determine community patterns; 2. Environmental changes can influence distribution patterns and relationships between species; 3. The co-occurrence relationship between species; 4. The History of Booderee National Park and 5. The Manuscript Investigation.

Line 121 - This paragraph is written in a derogatory manner. I suggest removing

Line 133 - Quantify (in percentage for example) the diet overlap of these species, as this will give you a background with data on food niche overlap.

Line 142 - Show a satellite map with the location of the study area, so it is possible to exemplify how the local landscape is and, if possible, identify the areas that suffered fires.

Line 152 - Make a collection scheme, a demonstrative figure is better to understand what was done.

Line 153 - Clarify the sampling independence of your traps, did each site have a number of traps? Was a distance between sites and between traps manipulated in relation to the home range of the animals? Only the home range of the vole is described.

Line 163 - Is it not clear how these variables were treated? I suggest seeing Zuur et al. 2010 and clarify.

Zuur, A. F., Ieno, E. N., & Elphick, C. S. (2010). A protocol for data exploration to avoid common statistical problems. Methods in ecology and evolution, 1(1), 3-14.

Line 217 - With the possibility of extracting new variables, I look forward to the new results.

Line 222- A graph showing the effect of each variable would be interesting.

Line 251 - Would these results then demonstrate something to be evaluated in the principle of competitive exclusion? It would be interesting to test.

Line 256 - Explore further the question of the effect of fire on the small mammal community.

Line 295 - No example found for mammals?

Line 303 - See here, an assumption, which has not been tested, so I suggest revising the term "introduced predator". "The increase in skunk abundance is supposed to be a response to the loss of Red fox".

Reviewer #2: First of all, I congratulate the authors for their efforts in producing this paper. I have just a few questions that I would like to see added to the text.

Has the introduction of exotic species brought any documented impact on the local fauna? This type of information would be interesting to enrich the discussion, even if it is not the objective of the work itself. Complementing the information on lines 298-300.

I missed an approach to relate environmental variables influencing other species, or works that have found environmental variables explaining the abundance of other species

Reviewer #3: This interesting study intends to assess the outcomes of a 20 years-old invasive predator control management program in an Australian National Park. The populational control of the red fox is correlated with an abundance increase of a local possum species, while the same is not observed for a bush rat species, whose abundance appears to be in decline. Aiming to understand this scenario, the authors analyze an impressive 14-year dataset of small mammal surveys and discuss the abundance patterns found for both these species within the theoretical grounds of niche occupancy and competition. The manuscript is generally well written, but although the data used and the statistical approach adopted to test the proposed hypotheses of the study seem sound, the way they are described in the Statistical Analysis section and the description of the hypotheses themselves are somewhat difficult to grasp and understand the rationale used. It took work, for example, to understand all model constructions within the modelling of each species, why they were different between species and how they relate to the hypotheses/prediction. Thus, it was not easy to follow how the authors used the results to support all the points raised in the discussion section. Given these observations, I listed some comments below that must be addressed before the study can be recommended to be published in PLOS ONE. That said, I currently recommend a major revision based on these commentaries.

Line 111 to 120: The authors mention they test “several predictions about species responses”, but they cite just one prediction regarding an inverse relationship between the abundance of both focal species. Then they list three hypotheses relating the effects of wildfire to the co-occurrence between both species. This whole construction seems counter-intuitive to me, as the three hypotheses are not all related to wildfires, and they are mentioned after a single prediction regarding the association between the abundances of the focal species. Additionally, there are no constructions related to the effects of fox control and no justifications for the listed hypothesis. What I would suggest here is to construct a single general hypothesis justified by the literature and based on the following questions:

- What is the expected effect of controlling foxes on the abundance of both possum and bush rat species? How might this be disturbed by the recurrent fire disturbances this National Park is subject to?

Then, devise predictions based on this general hypothesis, each addressing the topics previously cited (abundance respond to environmental variability? intraspecific competition? Which should be more important for each species)

Later in the manuscript, I would expect to see model selection routines based on each of these predictions.

Lines 121-122: What drives the authors to expect that? There needs to be a citation.

Also, how does this impact the hypothesis?

Line 125: Why is this section “Background” here? Not sure if it conforms with PLOS ONE´s article structure. Shouldn´t this information be in the introduction, helping the construction of hypotheses?

Line 165-166: What proportion of sites/years were not surveyed? It is essential to mention this here.

Line 172: Under this section, it is difficult to understand the description of the analytical procedures executed. In my opinion, this difficulty arises because the authors repeat a lot of the word “model”, confounding descriptions of the models used with descriptions regarding modelling construction/selection procedures. The authors need to improve the wording in this section and, maybe, produce a flow chart to aid comprehension. Some suggestions (if I managed to understand what steps were taken) :

- Line 173: “We used Bayesian regression models with a hurdles step (…) ” (and proceed in this paragraph to describe the models used). Additionally, please inform details regarding Markov Chains and methods to assess model convergence

- Line 179: “We performed two model selection procedures, one for each species (...)” (and then, proceed in his paragraph explaining that selection of the best models is based on several models constructed for each species based on the revised predictions of the introduction)

- Line xxx: Proceed with a paragraph to better explain (use citations) the model selection procedures, explain the use of Leave-on-out Cross Validation and how it optimizes the most parsimonious models by accessing the difference in log-likelihood of parameter estimates between them. Did you consider Multi-model selection (senso Burnham & Anderson, 2002; https://doi.org/10.1007/b97636)? If so, please state which delta-LOOIC interval was considered. Also, please see the following comments.

Lines 179-184: Why species followed different modelling approaches? Is this justified anywhere in the text?

Lines 190-194: Why were there no NULL models within the models constructed for each species?

Lines 2014-2015: Considering the Bayesian approach adopted, did you use the “expected log pointwise predictive density - elpd” of each model? Please clarify this point here.

Line 276-279: Given that you had models including the abundance of possums as a covariate for the model selection procedure for bush rats, but the inverse did not happen (abundance of bush rats as a covariate for possums), do you think that this could have an impact on this sentence “Our results demonstrate that an increase in the abundance of common brushtail possums was associated with a reduction in bush rat abundance.“?

Line 328-329: Blanchet, Cazelles and Gravel (2020) argue in a published paper in Ecology Letters (doi:10.1111/ele.13525) that when environmental conditions influence the distribution of two species, this could produce co-occurrence signals that could be misinterpreted as (positive or negative) ecological interactions. Given the correlative nature of your study and the fact that environmental variables are present in all models tested, would this not be an alternative explanation to the results obtained in this study?

6. PLOS authors have the option to publish the peer review history of their article (what does this mean?). If published, this will include your full peer review and any attached files.

Reviewer #1: No

Reviewer #2: No

Reviewer #3: No

---

## [Author Response · Author response to Decision Letter 0]

3 Aug 2023

Journal Requirements:

• Response: We have updated the formatting to meet PLOS ONE’s style requirements.

• Response: The data has been made available on the Dryad data repository; this has been updated within the Data availability section of the manuscript. (doi:10.5061/dryad.gmsbcc2sh)

3. We note that Figure 1 in your submission contain map images which may be copyrighted. All PLOS content is published under the Creative Commons Attribution License (CC BY 4.0), which means that the manuscript, images, and Supporting Information files will be freely available online, and any third party is permitted to access, download, copy, distribute, and use these materials in any way, even commercially, with proper attribution. For these reasons, we cannot publish previously copyrighted maps or satellite images created using proprietary data, such as Google software (Google Maps, Street View, and Earth). For more information, see our copyright guidelines: http://journals.plos.org/plosone/s/licenses-and-copyright.

• Response: We have included a satellite map using an ArcGIS Pro basemap, provided by Esri, an open-source mapping tool. We have included all the copyright and source details within both the figure caption and on the map itself, based on Esri’s requirements.

Reviewer #1 Comments:

Considering the manuscript, I believe the main potential would be how environmental and perturbation variables modulate the co-occurrence and abundance of opossum and rats species. I therefore ask reviewers to first review a new strategy in using the “introduced predator” question, what is discussed in the article are environmental factors influencing species co-occurrence patterns, but I do not see clarity on how the “introduced predator” factor is affecting or influencing the results. I believe that this is a determining factor within the context of the research, but I do not see it being tested or considered in the analyses, I only see it being mentioned.

I note that there was a very intense sample collection effort and I agree with the authors that more information is needed on the interactions between species and environmental variables. Therefore I suggest a new analysis taking into account the landscape of the study area. See the possibility of using images or a landscape database to use a delimitation in your sampling sites (buffer? investigating a minimum area that can be related to the home range of the species in focus), and extract information about this delimitation that may be fundamental in the occurrence of species (percentage of forested area, other natural conformations, distance from water bodies, other classifications). If the delimited areas overlap, randomize the points. Perform correlation analysis between variables and build models to explain how environmental variables and disturbance affect species abundance and co-occurrence. After the modifications, I suggest reassessing the data analysis. I suggest that a table be made in supplementary material with all collection points including values for each tested variable.

• Response: Thank you for raising an interesting point, however, after investigating some potential solutions based on your suggestions, we have decided to not incorporate any further analyses. We investigated these solutions by testing a selected range of landscape variables based on the habitat types within the park (e.g., forest, heathland, shrubland) at 200 m and 500 m buffers against bush rat abundance and ran a model selection comparing these models with a null model and the best fitting bush rat model. These landscape variables were found to be a poor fit, and had no effect on bush rat abundance, therefore we decided not to include these variables within our existing models.

Line 49 - Introduction: View a concept map strategy for getting started. 1. Environmental heterogeneity and factors that determine community patterns; 2. Environmental changes can influence distribution patterns and relationships between species; 3. The co-occurrence relationship between species; 4. The History of Booderee National Park and 5. The Manuscript Investigation.

• Response: Based on your suggestion, we have created a simple concept map to demonstrate our predicted ecosystem and the interactions within it, and placed it near the end of our introduction (Fig 1, Lines 113-116).

Line 121 - This paragraph is written in a derogatory manner. I suggest removing

• Response: We found this comment to be unclear, but we have made what we view as appropriate changes. We have moved the first sentence of this paragraph to the Discussion, where it is more relevant (Lines 331-334). We have also rewritten the second sentence, but have left it where it was, because we deemed it important for readers to understand this at this point (Lines 128-130).

Line 133 - Quantify (in percentage for example) the diet overlap of these species, as this will give you a background with data on food niche overlap.

• Response: We agree that the percentage of overlap would be interesting, we have not included this information as no study on this has been completed. We are, however, currently working on a comparative diet analysis, but do not have the results yet.

Line 142 - Show a satellite map with the location of the study area, so it is possible to exemplify how the local landscape is and, if possible, identify the areas that suffered fires.

• Response: We have changed the map to a satellite image, and included the areas burnt by fire since 2002 (Fig 2, Lines 170-176).

Line 152 - Make a collection scheme, a demonstrative figure is better to understand what was done.

• Response: We were unable to create a figure that would clearly explain the collection process, and therefore have focussed on improving the clarity within our text for explaining data collection (Lines 162-169).

Line 153 - Clarify the sampling independence of your traps, did each site have a number of traps? Was a distance between sites and between traps manipulated in relation to the home range of the animals? Only the home range of the vole is described.

• Response: We have revised the text to add more details about the trapping set up, including the number of traps and length of the transects, all of which were consistent for all of the sites (Lines 163-166). As the data were part of a long-term monitoring program, the site selection did not take specific species information into account. We also added information on the home range of common brushtail possums, with appropriate reference (Lines 144-146).

Line 163 - Is it not clear how these variables were treated? I suggest seeing Zuur et al. 2010 and clarify.

Zuur, A. F., Ieno, E. N., & Elphick, C. S. (2010). A protocol for data exploration to avoid common statistical problems. Methods in ecology and evolution, 1(1), 3-14.

• Response: We have improved the clarity of this text by adding more detail about data treatment after the description of the collection of the data for these variables. We transformed the vegetation and rainfall variables into quadratic functions using the poly() function (Lines 181-185).

Line 217 - With the possibility of extracting new variables, I look forward to the new results.

• Response: As seen in our response to your major issue, after careful consideration, we did not include the additional variables, and therefore the results have not changed.

Line 222- A graph showing the effect of each variable would be interesting.

• Response: We agree, and as such we have created forest plots for the best fitting models for each species, these have been placed in the supplementary information.

Line 251 - Would these results then demonstrate something to be evaluated in the principle of competitive exclusion? It would be interesting to test.

• Response: Based on the structure of our model and the limited ability of co-occurrence models to make causative statements, we were unable to assess the probability of the absence of bush rats, based on the presence of possums, therefore this study cannot comment on competitive exclusion. However, we will be performing other experiments that will assess whether this principle could apply.

Line 256 - Explore further the question of the effect of fire on the small mammal community.

• Response: We have reflected this comment by adding more information in the Discussion section on how to address the effect of fire (Lines 323-326).

Line 295 - No example found for mammals?

• Response: While we agree that this is a valid point, we have not changed this reference, as we are trying to generalise our results beyond mammals and Australia.

Line 303 - See here, an assumption, which has not been tested, so I suggest revising the term "introduced predator". "The increase in skunk abundance is supposed to be a response to the loss of Red fox".

• Response: We found this comment was unclear, but we have revised this sentence, including adding the term “introduced predator” (Lines 353-354).

Reviewer #2 comments:

Has the introduction of exotic species brought any documented impact on the local fauna? This type of information would be interesting to enrich the discussion, even if it is not the objective of the work itself. Complementing the information on lines 298-300.

• Response: We agree with this comment, and have added in a statement with references that will complement this information (Lines 347-349).

I missed an approach to relate environmental variables influencing other species, or works that have found environmental variables explaining the abundance of other species

• Response: To make this clearer, we have added more detail to explain this in the Introduction (Line 69).

Reviewer #3 comments:

Line 111 to 120: The authors mention they test “several predictions about species responses”, but they cite just one prediction regarding an inverse relationship between the abundance of both focal species. Then they list three hypotheses relating the effects of wildfire to the co-occurrence between both species. This whole construction seems counter-intuitive to me, as the three hypotheses are not all related to wildfires, and they are mentioned after a single prediction regarding the association between the abundances of the focal species. Additionally, there are no constructions related to the effects of fox control and no justifications for the listed hypothesis. What I would suggest here is to construct a single general hypothesis justified by the literature and based on the following questions:

- What is the expected effect of controlling foxes on the abundance of both possum and bush rat species? How might this be disturbed by the recurrent fire disturbances this National Park is subject to?

Then, devise predictions based on this general hypothesis, each addressing the topics previously cited (abundance respond to environmental variability? intraspecific competition? Which should be more important for each species)

Later in the manuscript, I would expect to see model selection routines based on each of these predictions.

• Response: We have re-ordered this paragraph and rewritten our overarching question, as well as hypotheses 1 and 3, to improve clarity (Lines 117-127). Further down, we have included a clearer explanation of our model selection methods, and made improvements to our Discussion section, to ensure all hypotheses are being accurately addressed.

Lines 121-122: What drives the authors to expect that? There needs to be a citation.

Also, how does this impact the hypothesis?

• Response: We expect competition rather than predation, as competitive behaviour of possums has been recorded more than predatory, but we do realise that predation is a potential interaction in this case. We have moved this sentence to the discussion, based on a comment from reviewer #1, and added references (Lines 331-334). Overall, we would likely see the same trends as we predict, regardless of the type of interaction.

Line 125: Why is this section “Background” here? Not sure if it conforms with PLOS ONE´s article structure. Shouldn´t this information be in the introduction, helping the construction of hypotheses?

• Response: On reflection we agree that this format did not make sense in its initial form. We have now moved this section to the Methods, and have renamed it ‘Species Information’ (Lines 137-149). We did this because the information at this level of detail is still important, but was too detailed to fit in before the aims, as we are keeping the introduction relatively broad.

Line 165-166: What proportion of sites/years were not surveyed? It is essential to mention this here.

• Response: We have now rewritten this sentence to be clearer that half of the sites were surveyed each year, depending on weather conditions (Lines 166-168). The surveys were conducted every year.

Line 172: Under this section, it is difficult to understand the description of the analytical procedures executed. In my opinion, this difficulty arises because the authors repeat a lot of the word “model”, confounding descriptions of the models used with descriptions regarding modelling construction/selection procedures. The authors need to improve the wording in this section and, maybe, produce a flow chart to aid comprehension.

• Response: We agree this section requires more clarity. As such we have rewritten the statistical analysis section of the Methods based on these notes (Lines 191-246).

Some suggestions (if I managed to understand what steps were taken) :

- Line 173: “We used Bayesian regression models with a hurdles step (…) ” (and proceed in this paragraph to describe the models used). Additionally, please inform details regarding Markov Chains and methods to assess model convergence

• Response: Thank you for these suggestions for ordering these paragraphs. We have added more detail to the description of the Bayesian regression models, as per the suggestions. We have reordered these paragraphs by splitting the information into two paragraphs, the first explaining the Bayesian regression models, and the second explaining the variables inputted into the models (Lines 192-214).

- Line 179: “We performed two model selection procedures, one for each species (...)” (and then, proceed in his paragraph explaining that selection of the best models is based on several models constructed for each species based on the revised predictions of the introduction)

• Response: We have now added more explanation to how we conducted the model selection process (Lines 215-227).

- Line xxx: Proceed with a paragraph to better explain (use citations) the model selection procedures, explain the use of Leave-on-out Cross Validation and how it optimizes the most parsimonious models by accessing the difference in log-likelihood of parameter estimates between them. Did you consider Multi-model selection (senso Burnham & Anderson, 2002; https://doi.org/10.1007/b97636)? If so, please state which delta-LOOIC interval was considered. Also, please see the following comments.

• Response: We put a lot of consideration into the selection of which method, before settling on LOOIC, this has now been explained in text (Lines 223-227). The delta-LOOIC was the difference between the best fitting model and each subsequent model, this was specified in the table captions.

Lines 179-184: Why species followed different modelling approaches? Is this justified anywhere in the text?

• Response: You have raised an important question. The main difference between the bush rat and common brushtail possum models is a control variable that is accounting for survey effort. We used different variables for the two species because while all trap types were present, bush rats were more likely to be captured in an Elliott trap, and possums were captured only in cage traps. We have now made this clearer within the text (Lines 210-214).

The other difference was that originally, bush rats weren’t inputted into the possum model as we were predicting a uni-directional relationships (i.e., we don’t expect bush rats to explain the changes in possum abundance), however, we have now added bush rats into the possum models, as these results should (and do) reflect what is occurring in the bush rat models.

Lines 190-194: Why were there no NULL models within the models constructed for each species?

• Response: Thank you for this point, this was an oversight on our part. We have now included the NULL models, which has confirmed our original model selection.

Lines 2014-2015: Considering the Bayesian approach adopted, did you use the “expected log pointwise predictive density - elpd” of each model? Please clarify this point here.

• Response: From our understanding, LOOIC uses a variation of the elpd equation within its methods, this has been added into the explanation of LOOIC.

Line 276-279: Given that you had models including the abundance of possums as a covariate for the model selection procedure for bush rats, but the inverse did not happen (abundance of bush rats as a covariate for possums), do you think that this could have an impact on this sentence “Our results demonstrate that an increase in the abundance of common brushtail possums was associated with a reduction in bush rat abundance.“?

• Response: We have now added bush rats into the possum models, and have found an equivalent result in the inverse relationship.

When we were forming the second hypothesis, and therefore the appropriate models, we were considering the potential effect of possums on bush rats, and as this method is correlative, we expected that the same relationship would be observed if the bush rats and possums were switched. Therefore, this sentence was an accurate comment of the relationship we are seeing, when predicting the direction of the relationship.

Line 328-329: Blanchet, Cazelles and Gravel (2020) argue in a published paper in Ecology Letters (doi:10.1111/ele.13525) that when environmental conditions influence the distribution of two species, this could produce co-occurrence signals that could be misinterpreted as (positive or negative) ecological interactions. Given the correlative nature of your study and the fact that environmental variables are present in all models tested, would this not be an alternative explanation to the results obtained in this study?

• Response: Thank you for this insight, we have added this reference into the discussion (Lines 317-320) to make the limitations of our model clearer, and then referred back to this point within the conclusion (Lines 381-382).

---

## [Decision Letter · Decision Letter 1]

14 Sep 2023

PONE-D-23-02032R1Environmental variables influence patterns of mammal co-occurrence following introduced predator controlPLOS ONE

Dear Dr. Kanishka,

Thank you for submitting your manuscript to PLOS ONE. After careful consideration, we feel that it has merit but does not fully meet PLOS ONE’s publication criteria as it currently stands. Therefore, we invite you to submit a revised version of the manuscript that addresses the points raised during the review process.

We look forward to receiving your revised manuscript.

Kind regards,

Daniel de Paiva Silva, Ph.D.

Academic Editor

PLOS ONE

Journal Requirements:

Additional Editor Comments (if provided):

Dear Dr. Kanishka,

After this second review round, you may consider that your manuscript has been accepted for publication (in practice) in PLoS One. Still, you will see that one of the reviewers indicated that minor reviews (your dataset deposited in Dryad) are still needed. Therefore, as soon as you proceed with the minor improvement suggested by the reviewer, your manuscript will be formally accepted for publication in PLoS One.

Best regards,

Daniel Silva

Reviewers' comments:

Reviewer's Responses to Questions

**Comments to the Author**

1. If the authors have adequately addressed your comments raised in a previous round of review and you feel that this manuscript is now acceptable for publication, you may indicate that here to bypass the “Comments to the Author” section, enter your conflict of interest statement in the “Confidential to Editor” section, and submit your "Accept" recommendation.

Reviewer #1: All comments have been addressed

Reviewer #2: All comments have been addressed

Reviewer #3: All comments have been addressed

2. Is the manuscript technically sound, and do the data support the conclusions?

Reviewer #1: Yes

Reviewer #2: Yes

Reviewer #3: Yes

3. Has the statistical analysis been performed appropriately and rigorously? 

Reviewer #1: Yes

Reviewer #2: Yes

Reviewer #3: Yes

4. Have the authors made all data underlying the findings in their manuscript fully available?

Reviewer #1: Yes

Reviewer #2: Yes

Reviewer #3: Yes

5. Is the manuscript presented in an intelligible fashion and written in standard English?

Reviewer #1: Yes

Reviewer #2: Yes

Reviewer #3: Yes

6. Review Comments to the Author

Reviewer #1: All points raised about review were answered coherently. Therefore, I believe in the potential of the manuscript.

Reviewer #2: The authors will answer all the questions and doubts, to figure out now very best and most informative about the study area. I am grateful for the responses and parabenization of the authors for their work.

Reviewer #3: The authors addressed all my previous commentaries and significantly improved the clarity of their analytical approach, which was my main concern. This improvement, in turn, made their conclusions well-supported, and I congratulate the authors for their nice work. Before fully accepting the manuscript, my last remark is that their data still needs to be available on Dryad, as stated in the manuscript. Thus, I could not check if, together with the data, the author also made R scripts of their analytical approach available. If not, I encourage the authors to do so and fully contribute to reproducibility and transparency in science.

7. PLOS authors have the option to publish the peer review history of their article (what does this mean?). If published, this will include your full peer review and any attached files.

Reviewer #1: No

Reviewer #2: No

Reviewer #3: No

---

## [Author Response · Author response to Decision Letter 1]

28 Sep 2023

Reviewer #1: All points raised about review were answered coherently. Therefore, I believe in the potential of the manuscript.

Thank you for your support in our manuscript.

Reviewer #2: The authors will answer all the questions and doubts, to figure out now very best and most informative about the study area. I am grateful for the responses and parabenization of the authors for their work.

We are grateful that we have addressed all questions. Thank you for your support in our manuscript. 

Reviewer #3: The authors addressed all my previous commentaries and significantly improved the clarity of their analytical approach, which was my main concern. This improvement, in turn, made their conclusions well-supported, and I congratulate the authors for their nice work. Before fully accepting the manuscript, my last remark is that their data still needs to be available on Dryad, as stated in the manuscript. Thus, I could not check if, together with the data, the author also made R scripts of their analytical approach available. If not, I encourage the authors to do so and fully contribute to reproducibility and transparency in science.

Thank you for your comments, and the advice given in the last round of revisions. We have now made our data public on the Dryad digital repository (link: https://doi.org/doi:10.5061/dryad.gmsbcc2sh.). Our software (i.e., R code) has also been made publicly available (link: https://doi.org/10.5281/zenodo.8378280).

---

## [Editor Report · Decision Letter 2]

3 Oct 2023

Environmental variables influence patterns of mammal co-occurrence following introduced predator control

PONE-D-23-02032R2

Dear Dr. Kanishka,

We’re pleased to inform you that your manuscript has been judged scientifically suitable for publication and will be formally accepted for publication once it meets all outstanding technical requirements.

Kind regards,

Daniel de Paiva Silva, Ph.D.

Academic Editor

PLOS ONE
---

## [Editor Report · Acceptance letter]

8 Oct 2023

PONE-D-23-02032R2 

Environmental variables influence patterns of mammal co-occurrence following introduced predator control 

Dear Dr. Kanishka:

I'm pleased to inform you that your manuscript has been deemed suitable for publication in PLOS ONE. Congratulations! Your manuscript is now with our production department. 

Kind regards, 

on behalf of

Dr. Daniel de Paiva Silva 

Academic Editor

PLOS ONE